# 'Maze' not pathway: focus group exploration of patients' and public experiences of the UK NHS elective total joint arthroplasty pathway

Sarah Jasim [ORCID],[1] Fiona Aspinal [ORCID],[2] Raj Mehta,[2] Jean Ledger,[2] Rosalind Raine,[2] Naomi J Fulop [ORCID],[2] Helen Barratt [ORCID] [2]

[1]Care Policy and Evaluation Centre, The London School of Economics and Political Science, London, UK
[2]Department of Applied Health Research, University College London, London, UK

**Correspondence to**
Dr Sarah Jasim;
s.jasim@lse.ac.uk

## ABSTRACT

**Objective** To explore patient and public perceptions of planned improvements to the National Health Service (NHS) total joint arthroplasty (TJA) pathway.

**Design** Three qualitative focus groups undertaken March–May 2018, as part of a mixed-methods evaluation of Getting It Right First Time. Transcripts were subject to framework analysis to identify thematic content between October 2018 and October 2021.

**Setting** Elective TJA surgery in the English NHS.

**Participants** Two focus groups including patients who had undergone TJA in the previous 2 years (group 1: n=5; group 2: n=4) and the other individuals who had not but were aged 60+ (group 3: n=5). Participants were recruited via community groups and patient panels.

**Results** Fourteen individuals took part in the focus groups; all were aged over 60; seven (50%) were female and nine (64%) had undergone TJA surgery. Participants' perspectives were categorised into themes and mapped onto stages of the TJA pathway. Although perioperative care is often the focus of improvement efforts, participants argued that the patient journey begins before individuals present to primary care. Participants had concerns about other aspects of the pathway, such as obtaining a surgical referral, with prereferral interventions aimed at potentially avoiding the need for surgery (ie, physiotherapy) being perceived as a mechanism to restrict access to secondary care. Patient experience was also conceptualised as a 'maze', rather than the logical, sequential process set out in clinical guidelines; exacerbated by a lack of information, knowledge and power imbalances.

**Conclusion** The linear conceptualisation of the TJA pathway is at odds with patient experience. Improvement programmes focused on perioperative care fail to consider patient concerns and priorities. Patients should be directly involved in improvement programmes, to ensure that patient experience is optimised, as well as informing related processes and important outcomes of care.

---

## STRENGTHS AND LIMITATIONS OF THIS STUDY

⇒ This is the first study to look at the entire total joint arthroplasty pathway from the perspective of patients and the public and explore how their priorities differ from those of clinicians and managers seeking to improve care.

⇒ Coproducing research with a patient advisor ensured the patient perspective was central to our work from planning through to analysis and dissemination.

⇒ We sought to recruit via a wide range of different channels, including professional societies, patient groups and local community groups (including multifaith organisations).

⇒ Nevertheless, we were not able to recruit a diverse sample with respect to ethnicity, and one participant withdrew because of fears of cultural stigma.

⇒ As participants volunteered to participate, there may be a degree of self-selection bias, with participants who were more confident in articulating their views and experiences or indeed, those with particular types of experiences, potentially more likely to participate.

---

an increasingly ageing population, and trend for TJA earlier in life,[4] the predicted growth in global demand is likely to have a significant impact on health systems,[5] such as the National Health Service (NHS) in England.

A care pathway[6] is a multidisciplinary tool for planning the care of specific, well-defined groups of patients who have a predictable clinical course, such as TJA. Drawing on research evidence and best practice, it sets out how different components of care should be optimised and sequenced, to enhance quality[7] and improve patient safety. This approach has been shown to reduce unnecessary variation[6 8–11] and may improve patient outcomes and optimise resource use.[6 12–14] Standardisation of care is important for national improvement programmes such as the English 'Getting It Right First Time' programme

## BACKGROUND

Elective hip and knee replacement are the two most common total joint arthroplasty (TJA) procedures, and two of the most frequently performed and effective surgical procedures in the UK[1 2] and globally.[3] With

(GIRFT), which aims to tackle unwarranted variation in orthopaedic surgery across the NHS.[15 16] GIRFT sought to develop a 'best of the best' clinical pathway[17] to improve patient experience and expedite recovery following TJA. Although clinical perspectives are central to pathway development,[8] the views of current and future patients who must navigate them, are equally important to optimise experience, outcomes and efficiency.[18]

In the UK, general practitioners (GPs) act as gatekeepers to secondary care, including the TJA pathway.[19] Patients experiencing hip or knee pain typically first present to a GP. In some English regions, local commissioners (leaders responsible for purchasing healthcare on behalf of local communities) are increasingly setting criteria that patients must meet before they can be referred to an orthopaedic surgeon. Current guidance recommends conservative management initially, for at least 3 months, with the aim of potentially avoiding the need for the patient to undergo surgery. Interventions include medication, physiotherapy and/or support with lifestyle and weight loss. If this is not successful, the GP should refer the patient to an orthopaedic surgeon to discuss potential surgery.

Existing research about patient perspectives of the TJA pathway has tended to focus on patient satisfaction and experience[20] at specific stages of the pathway, such as deciding to undergo surgery,[21] preoperative education,[22] waiting for surgery[23 24] and management of postoperative pain.[12] Given shared decision-making is considered integral to care pathways as well as wider efforts to improve care, there is currently no evidence on how patients and public view planned improvements to the TJA pathway, and whether their priorities align with those of clinicians and managers.[25 26] This study sought to address that gap by exploring patient and public perceptions of planned improvements to the TJA pathway, including the factors that influenced this. The research was part of a wider mixed-methods evaluation of the GIRFT programme.[15 16 27]

## METHODS
### Study design and recruitment
Focus groups enabled us to collect views from multiple participants simultaneously and explore meanings behind those views, which were generated, discussed and refined through group interactions in a way that could not happen with other methods.[28]

To explore the outcome of interest, which was patient and public perceptions of planned improvements to the NHS TJA pathway, we sought to recruit between four and eight participants to each of three focus groups. This was based on evidence regarding the optimal number of groups and participants for focus group research.[29] Groups 1 and 2 included patients who had undergone an elective hip or knee replacement in the previous 2 years (March 2016–March 2018). Group 3 included individuals who had not undergone TJA but were in the age bracket most likely to receive this type of intervention (age 60+) (figure 1).[30] This enabled us to compare experiences of those who had undergone TJA with the expectations of those who had not, and how this influenced participants' views. In this paper, we report the findings from all three groups together, referring to individuals as participants, except where there were clear differences between those who had undergone surgery, and those who had not. We recruited participants via community organisations across London (UK) whose membership was likely to be aged 60 and over (eg, third sector organisations, day centres and activity clubs). We also invited participation from eligible patient advisory group members from relevant organisations (eg, the British Orthopaedic Association). Researchers (SJ and JL) conducted telephone and email

---

**All groups:**
- Understand and speak English well enough to be able to give informed consent and participate in a group discussion
- Cognitively able to give informed consent and participate in a group discussion
- Not on a waiting list for total joint arthroplasty (TJA) (in case discussions affect their willingness to undertake TJA surgery)
- Aged over 18 years

**Patient groups:**
- Undergone elective TKA or THA between March 2016 and March 2018 (to minimise recall bias)
- Received **only** unplanned THA or TKA (because this is a different pathway)

**Public group:**
- Aged 60 years or older
- have **not** had TJA, but are willing to discuss NHS services to treat joint problems from a public perspective

**Figure 1** Eligibility criteria.

| Table 1 | Participant characteristics | | |
|---|---|---|---|
| | Group 1 (patients— n=5) | Group 2 (patients— n=4) | Group 3 (public—n=5) |
| Sex | Male (n=2), female (n=3) | Male (n=2), female (n=2) | Male (n=3), female (n=2) |
| Age range | 60–69 (n=2) 70–79 (n=3) | 60–69 (n=1) 70–79 (n=1) 80–89 (n=2) | 60–69 (n=3) 70–79 (n=2) |
| Ethnicity | White British (n=4) White European (n=1) | White British (n=3) White South African (n=1) | White British (n=4) White North American (n=1) |

screening and collected preliminary demographic information to maximise sample diversity (table 1).

## Data collection

We conducted three focus groups between March and May 2018 in an academic (non-healthcare) setting. Observation data were collected by senior researchers (FA or HB) to capture any contextual factors that might have impacted data quality, such as the topics being discussed or depth of discussion. All participants gave informed consent. Each focus group began with a brief presentation by the research team, providing a high-level overview of efforts to improve the TJA pathway for patients, including GIRFT.[15]

Focus group discussions were informed by semistructured topic guides (online supplemental file 1 and 2), which included participants' priorities before, during and after surgery, as well as their views of the planned improvements to the TJA pathway. Each group lasted approximately 2 hours and was audiorecorded using an encrypted device for professional transcription in full.

## Patient and public involvement

Three focus groups were cofacilitated by a researcher (SJ or JL) and the patient advisor (RM).[31] Following advice from our patient advisor (RM),[31] we (SJ) also provided participants with a simplified illustration of the UK TJA care pathway (figure 2), based on clinical guidance at the time.[32] The pathway diagram and topic guides were piloted with a research advisory panel of patient advisers and refined initially, and then iteratively as the research progressed (ie, diagram title and emphasis of questions), to take account of emerging findings, in line with accepted qualitative methods.[33] Preliminary results of the study were verbally presented to participating patient advisors.

## Data analysis

The core research team (HB, FA, RM, SJ and JL) held debrief meetings after each focus group to ensure triangulation between focus group findings and researcher observations. Observation notes and field notes were not subject to formal analysis but were used in debrief and analysis meetings to ensure transcript data were contextualised. Transcript data were analysed using a framework approach.[34] We used the care pathway diagram (figure 2) and structure of the topic guides (online supplemental file 1 and 2) to develop the initial coding framework, and

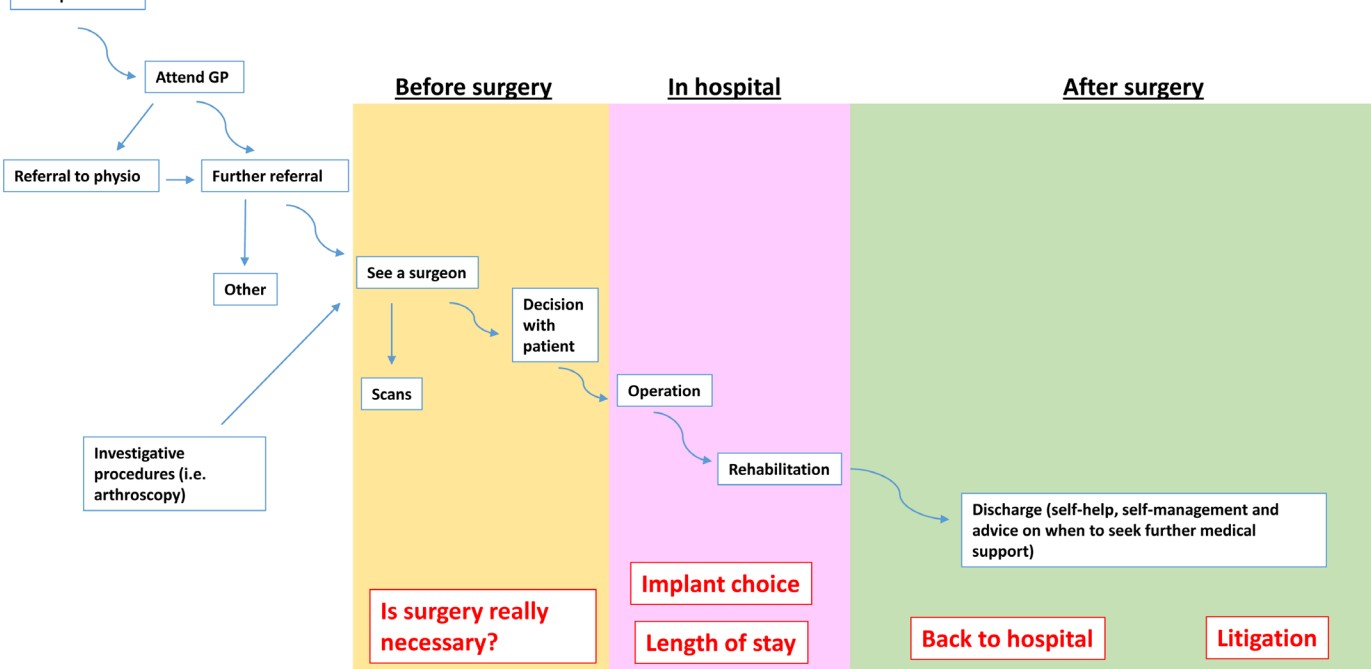

**Figure 2** Patient pathway diagram.

incorporated concepts and meanings identified in the data.[13]

The first stage of analysis involved familiarisation, where researchers (SJ, JL, FA and HB) read and collaboratively reviewed transcripts to understand meanings, concepts and ideas, in a reflexive manner (see online supplemental file 3). The wider research team (SJ, JL, RM, FA, HB and NJF) then identified and confirmed parameters of the main themes to form the basis of a thematic framework in a series of separate meetings. Data were indexed based on the thematic framework using MAXQDA software (SJ). Indexed data were organised into thematic charts for each category, in Microsoft Excel, using direct quotations where possible. These were reviewed and modified when necessary to clarify meaning (FA). Themes were further mapped and interpreted to understand areas of significance to patients and the public (SJ, FA, HB, JL, RM, NJF and RR).[15]

## RESULTS

### Participants

Seventeen individuals agreed to participate, although three withdrew on the day. Two because of heavy snowfall (group 2 and group 3) and one due to fears of cultural stigma from openly discussing experiences (group 2). All 14 participants were aged over 60 and all self-identified as white British or white Other. Seven were female, and nine participants had undergone TJA surgery in the previous 2 years. Of the five who had not undergone surgery (ie, group 3), three had friends or family who had previously undergone TJA.

Our original goal was to explore participants' views about planned improvements to the TJA pathway (such as GIRFT),[2 15] which were focused particularly on the perioperative 'in-hospital' (figure 2) phase of the care pathway. However, it quickly became apparent that other phases of the pathway were of equal, if not more, importance to participants. Participants understood the pathway diagram, but argued that while the streamlined, linear presentation may be how it appears to clinicians and managers, it did not reflect their experiences. Here, we report participants' views about priorities for pathway improvement; the realities of traversing the TJA pathway; and barriers and facilitators they encountered while navigating the pathway (participant quotations are provided in table 2).

### Patient priorities: obtaining a referral to see an orthopaedic surgeon

Although the perioperative phase of the pathway is often the focus of improvement programmes, such as GIRFT, individuals who had undergone TJA argued that obtaining a referral to an orthopaedic surgeon is in fact a greater challenge and concern (see table 2, quote 1). This was 'upsetting' given their level of pain or disability (quote 2) especially when appointments were postponed. Some participants found that GP-recommended non-surgical interventions resulted in little or no improvement, while others had been in too much pain to engage with physiotherapy (quote 3). Consequently, they viewed this approach simply as a mechanism for managing demand (quote 4). Some participants believed that GPs had pressure placed on them to not refer patients to secondary care (quote 5,). Others felt that external pressures to ration NHS services impacted their access, such as referrals being reviewed by an 'assessment committee' (quote 2) or appointments being delayed by repeated, unexplained cancellations.

Because of the obstacles they had faced, participants spoke about the difficulties encountered to get onto the pathway and secure an orthopaedic referral, and in some cases, 'persuade' (quote 6) or even 'bully' (quote 7) healthcare professionals, while on this conveyor belt' (quote 17) or 'production line' (quote 8).

### The realities of the elective TJA pathway

Participants, especially those who had not undergone surgery (group 3), emphasised the importance of joined-up care planning throughout the pathway, including clinical teams planning surgery dates to coincide with available postoperative support (quote 9). Disconnected communication and ways of working between health providers at different stages of the pathway, left patients (groups 1 and 2) having to assume responsibility for their own care, despite feeling powerless with insufficient information to make such informed decisions (quote 10). The lack of joined-up care was important to all groups and seen as being exacerbated by poor communication. For example, they described poor discharge procedures where communication with wider services was lacking, such as pharmacy, community health services, rehabilitative physiotherapy (quote 11) and social care (quote 12).

### Navigating the 'maze'

Without joined-up forward-planning, patients had to manage and navigate a range of complex local health systems alone. They were left feeling adrift from the system, and many participants felt they were ill-equipped to tackle the obstacles they encountered because of asymmetries of both information and power. As one neatly summarised, the pathway is 'like a maze' (quote 13).

#### Lack of information and knowledge

Participants reported that from the point of referral from primary care, they were 'left in the dark' (quote 14). A lack of communication (quote 15) or appropriate, timely information from their healthcare provider hindered their understanding of the process (quote 16) making it difficult for them to make informed choices . This was compounded for some by a lack of knowledge, meaning that they felt they did not know what questions they should be asking. This was particularly a concern in the stages prior to surgery but differed according to participants' direct or indirect experiences; location of care;

**Table 2** Focus group (FG) quotes

| Ref. | FG no. | Gender | Quote |
|---|---|---|---|
| Quote 1 | 1 | Male | '…… that's [pointing to the first part of the pathway diagram] probably where most problems, as far as I'm concerned, arose. In that trying to see a consultant was very difficult.' |
| Quote 2 | 1 | Male | '…I get a letter in the post saying, it's a confirmation of the date, so that would be followed by another letter that says they've just cancelled that appointment…With no explanation, it just says it's cancelled… "What happens, where are we now, or what happens now?"…they said, "Oh well, it's got to go in before the assessment committee"… Well, this is very, very … upsetting that… I'm in a lot of pain, I want to see a consultant, and there's all these obstacles in the way before you can even see the consultant.' |
| Quote 3 | 1 | Female | 'I had months of physiotherapy right through till February the following year. It wasn't helping, I didn't think it was helping a lot… And then in the February I spoke to my physiotherapist, asked him… Am I going to go on and on like this? … So, I'm now thinking, "…they're just… shuffling me along," obviously … would I like to just go ahead with the op…' |
| Quote 4 | 1 | Male | 'Yeah, I think it's a postcode lottery, isn't it, where some people live that there's a huge waiting list and I think the GPs…refer them to the physiotherapist first so… keeps them out of the way for a bit…' |
| Quote 5 | 2 | Male | '…the GP is the stopping point, and it causes a lot of the other things going up the line to be worse than they need to be because they resist. Obviously, they're the Clinical Commissioning Group that actually drives them on this and say, 'Don't refer, don't refer, don't refer, we'll even give you some extra bonus points if you don't refer.' |
| Quote 6 | 2 | Male | '…so it was down to you to look after your own, and when it all starts to go wrong, you've got to go to your GP and you've got to persuade him to refer you to a specialist…' |
| Quote 7 | 1 | Female | '…I felt I had to bully them in order to have the surgery…' |
| Quote 8 | 1 | Female | '– I was just taken aback by how much I felt like a widget on a production line.' |
| Quote 9 | 3 | Female | '…if you like, you'd need a referral to Social Services. I think that should be part of the pre-operative planning rather than suddenly realising afterwards. What I was talking about was the OT assessment and the follow-up physiotherapy, I wasn't talking about Social Services, about social care support, which you may well need and that should have really been part of the planning beforehand. Because if you've got to wait 'til afterwards you'll be in hospital for a month before anything happens. I know because I've been a social worker so, you know, I know what it's like.' |
| Quote 10 | 1 | Female | 'lack of communication between various people, the doctors, the nurses, so that I felt I was the person who knew more than anyone else about what was going on, but I knew least about what should be going on' |
| Quote 11 | 3 | Female | 'I'd like to know about physiotherapy and after care and the system they use…So I'd want to know, if I was going to go in, whether I'd be not only would the surgery be good and efficient but there would be a very good programme coming out in after care and who'd be doing that with me.' |
| Quote 12 | 3 | Female | 'I want to come back to the Social Services and what happens when you get out of hospital, especially if you live alone or… you know, how do you manage for that period of time after hospital, that's an important handover and that's not always handled really well.' |
| Quote 13 | 1 | Male | '…like a maze' |
| Quote 14 | 1 | Male | 'all the time there's never any communication as to what's happening. So, you're left in the dark, and that itself is upsetting because, you know, you want to get it dealt with, you don't know at what stage it's going to happen, and you're just being pushed from pillar to post, and nobody will give you any information.' |
| Quote 15 | 3 | Female | 'There's a big thing about sort of communication through the line so that people don't rely on you as the rather disorientated patient after an operation to actually be the person who's informing so that there are other people sharing information, but you've also got a voice, that your voice is going to be heard or that you've elected someone to speak for you and that person's voice is going to be heard as well.' |
| Quote 16 | 3 | Male | 'information tends to be… You're always given something when you go out—and this is all kinds of hospitalisation, not just surgery… but it's never quite enough… but it's generally not good…' |

Continued

| | | | |
|---|---|---|---|
| **Table 2** | Continued | | |
| **Ref.** | **FG no.** | **Gender** | **Quote** |
| Quote 17 | 1 | Female | 'because I can see from this chart [diagram] that I'm on a conveyor belt and that's the feeling that I had. And actually, I felt that I'd been in the dark from the start. And I felt that there is information there if I ask the right question, but I don't know what question to ask. And I have felt that I have to look it up a lot myself, I have to find out things myself' |
| Quote 18 | 2 | Female | 'Well, personally I wasn't aware that there was even any choice.' |
| Quote 19 | 3 | Female | 'It is actually a plus point for hospitals to make their statistics available publicly in some way because people will search for excellence and people will avoid mediocrity, and I certainly don't depend on my GP for that…' |
| Quote 20 | 1 | Female | 'So, that question of choice actually goes all the way through. Is it really choice or not and how much power do we really have? How much knowledge do we have?' |
| Quote 21 | 1 | Female | 'I was so intimidated by him, and so overwhelmed by him, I thought, 'Right, okay, well, you know, I'm just going to … there's no point in my doing anything else here'.' |
| Quote 22 | 3 | Female | '…care-givers are important too and I can't tell you how many times I was at a doctor and the doctor just ignored the questions that I had to ask because I was not the patient…' |
| Quote 23 | 1 | Female | 'But it was about two weeks before my operation, so if you are anxious and you feel like you're in the dark, that's a long time to wait before you get those questions, a chance to answer those questions, and you find out new information. It was at the joint school that I found out that you don't have a general anaesthetic, that you have a spinal anaesthetic.' |
| Quote 24 | 1 | Female | 'But then physio told me that there was a system whereby you get a choice of hospitals for having your operation in. I didn't know that either. She said, 'Haven't you heard about this before?' And she gave me a list of six hospitals, the physio… she said, 'Look them up, go and read the CQC report on these hospitals and see which one you like, and if you prefer one of the other ones, come back and let me know'. I'm sorry, having come through all this and been so in the dark I'm now being told to decide on what hospital.' |
| Quote 25 | 1 | Female | 'I knew the system more, I was much more able to work my way through it, to talk to the right people, to make the decision' |
| Quote 26 | 3 | Female | '…I'd want the occupational therapist to assess me, assess my home, work out what the … support I'm going to get at home to actually help me achieve mobility and how accessible they are…' |

and pre-existing knowledge or professional background (quote 17).

Although clinical guidance[35] implies shared decision-making should be built into the pathway, participants' knowledge about available choices varied. For example, some participants in all groups had not known that they could choose the hospital they went to, or their surgeon, while others had sought out information to inform their choice (quote 18, quote 19). Nevertheless, even those who were aware that they could exercise choice, felt that they lacked sufficient knowledge to do so (quote 20).

### Power imbalance

Alongside a lack of information and communication, many participants described how they had to disrupt existing power dynamics to navigate the TJA pathway. Asymmetries of power in relationships with key health professionals could significantly impact their experience of care (quote 8) and their progress along the pathway. For example, negative interactions with a surgeon had halted one participant's journey during their first appointment with a surgeon and hindered their access to pathway at the very start (quote 21). Another participant, who had

cared for a TJA patient, described how her attempts to advocate for them were ignored (quote 22).

### Overcoming obstacles

Participants described three distinct sources of knowledge which they drew on to differing degrees to overcome the obstacles they faced: formal learning, informal learning and lived experience.

### Formal learning

Formal learning opportunities offered to patients prior to surgery had helped some. For example, preoperative education groups (also known as 'Joint Schools') run by hospital staff for patients listed for surgery, had alleviated fears, increased knowledge and provided an opportunity to ask questions. Nevertheless, for some, the programme was offered very close to the date of the procedure, meaning that they had a long wait to raise their concerns. Consequently, some would have liked this to have been offered much earlier in the pathway (quote 23).

### Informal learning

Some participants also sought out or expected to seek out information for themselves that they thought they needed

to navigate the process. This included details about medication (carer, group 3); chasing appointments (female patient, group 1); and undertaking their own research to inform choices (all groups) and guide decision-making (female patient, group 1). Others had also sought information from health professionals such as physiotherapists who were familiar with the TJA pathway, which helped them feel 'less in the dark' (quote 24).

### Lived experience

Participants who were able to draw on a previous experience of undergoing surgery, used this to help inform choices and decision-making and felt better able to navigate the pathway on subsequent occasions. Those who had previously undergone a TJA knew to request additional information about regaining function, resuming activities and exercising, not only to improve outcomes but also to assist navigating the crucial postoperative stage of the pathway (quote 26). Some also believed that this knowledge had enabled them to expedite the process on subsequent occasions, including identifying potential short cuts (quote 25,).

## DISCUSSION

Despite ongoing efforts to streamline and standardise care, patient experience differs significantly from the seemingly logical, sequential process set out in clinical guidelines.[35] Importantly for patients, the pathway does not begin and end at the doors of the hospital. For example, it starts at the first presentation to primary care with joint pain. A lack of timely information and communication also leaves patients feeling that they have been 'left in the dark' to navigate a 'maze'. This sense is exacerbated by power and information asymmetries between patients and clinicians. While formal education programmes such as 'Joint School' can be helpful, many patients have to seek out additional information for themselves to answer their queries, understand the procedure and navigate the TJA pathway.

Previous research has largely focused on specific phases of the care pathway. This is the first study to look at the entire TJA pathway from the perspective of patients and the public and explore how their priorities differ from those of clinicians and managers seeking to improve care. Coproducing research with a patient advisor ensured the patient perspective was central to our work from planning through to analysis and dissemination. Qualitative data collection methods were used to allow participants to describe their experience and views in their own words, and the focus group approach enabled sharing, comparing and corroboration of ideas. As socioeconomic circumstances may impact a patient's decision to undergo TJA,[36] we sought to recruit via a wide range of different channels, including professional societies, patient groups and local community groups (including multifaith organisations). Nevertheless, we were not able to recruit a diverse sample with respect to ethnicity. Indeed, one

participant withdrew because of fears of cultural stigma. It is acknowledged that cultural differences may exist between other groups of patient and public members. The study methodology and number of study participants may limit the transferability of findings—although this was mitigated by using purposeful sampling there was geographical variability in participants' lived experiences of the UK TJA pathway.[37] The methodology was selected to explore a range of views in depth, and not necessarily representation. As participants volunteered to participate, there may be a degree of self-selection bias, with participants who were more confident in articulating their views and experiences or indeed, those with particular types of experiences, potentially more likely to participate.

Our findings have several implications for clinicians and policymakers. Although perioperative care is often the focus of clinically led efforts (eg, GIRFT)[2 27] to improve the TJA pathway, participants also emphasised that obtaining a referral to a surgeon is often the greatest challenge, and the postoperative period following surgery is a priority. Despite some evidence to suggest that prehabilitation is associated with improving some of the outcomes for patients undergoing orthopaedic surgery both preoperatively and postoperatively, we have highlighted that non-surgical interventions (eg, physiotherapy) prior to surgical referral are perceived by patients and the public as a means of managing demand.[38] While this remains the case, as well as potentially damaging patient trust, it is likely to impact patient engagement with evidence-based care. Efforts to provide patients with information about their procedure and likely recovery (eg, via 'Joint Schools') are welcome, but many want information about the pathway earlier than this so that they are not left feeling ill equipped to navigate the 'maze' of care themselves, especially if they have not previously undergone surgery. At a policy level, those working to standardise services using care pathways should also be aware that patient experience is seldom analogous to their linear, stepwise conceptualisations. In line with the existing literature, our findings suggest that experience-based codesign should be a core part of pathway redesign.[39] Challenges with traversing the TJA pathway, alongside the negative impacts on physical and mental well-being of delayed appointments that were reported in this study, are likely to be exacerbated by the COVID-19 pandemic which has resulted in further delays, cancellations, and increased waiting times for those on the TJA pathway.[40] The subsequent impacts on the health system have increased the length of patients' experience of pain, decreased mobility and worsened quality of life.[41 42]

## CONCLUSION

In conclusion, we have highlighted the disparity between the priorities of improvement programmes such as GIRFT and those of patients and the public. While improvement programmes typically seek to improve perioperative care, patient concerns lie elsewhere, for example, obtaining

a referral to an orthopaedic surgeon and managing recovery at home. Research that includes views from both patients and healthcare professionals is needed at these crucial stages of the TJA pathway to explore how patients are being supported. The typically linear conceptualisation of TJA pathways in policy documents is also at odds with how patients experience the process. With rates of TJA set to increase globally, patients and public must be directly involved in improvement programmes to optimise not only processes and outcomes of care, but also patient experience.

**Acknowledgements** The authors would like to acknowledge and thank the focus group participants, and especially Steven Towndrow, Wendy Chandler, Stephanie Hume, Patricia Hallam and the NIHR ARC (formerly CLAHRC) North Thames Research Advisory Panel patient advisers for their efforts in preparation and organisation of the focus groups.

**Contributors** HB was principal investigator. HB and RR initiated the research. HB, NJF and RR designed the evaluation. SJ, RM and JL were responsible for qualitative data collection; SJ, FA, RM, JL, NJF and HB analysed the qualitative data. SJ, FA, RM, JL, RR, NJF and HB drafted the manuscript and all authors contributed to reviewing and substantive revision. All authors approved the final version. All authors had full access to all the data in the study and accept responsibility to submit for publication. HB will act as guarantor.

**Funding** This study was funded by National Institute for Health and Care Research (NIHR) Programme Grants for Applied Research (Applied Research Collaboration (ARC) North Thames).

**Competing interests** None declared.

**Patient and public involvement** Patients and/or the public were involved in the design, or conduct, or reporting, or dissemination plans of this research. Refer to the Methods section for further details.

**Patient consent for publication** Not applicable.

**Ethics approval** This study involves human participants and was approved by NHS North West—Liverpool East Research Ethics Committee (16/NW/0654). Participants gave informed consent to participate in the study before taking part.

**Provenance and peer review** Not commissioned; externally peer reviewed.

**Data availability statement** No data are available.

**ORCID iDs**
Sarah Jasim http://orcid.org/0000-0003-3940-6350
Fiona Aspinal http://orcid.org/0000-0003-3170-7570
Naomi J Fulop http://orcid.org/0000-0001-5306-6140
Helen Barratt http://orcid.org/0000-0002-1387-137X

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
