## [Reviewer comments · BMJ Open]

ARTICLE DETAILS

TITLE (PROVISIONAL)	'Maze' not pathway: focus group exploration of patients' and public experiences of the UK NHS elective total joint arthroplasty pathway.
AUTHORS	Jasim, Sarah; Aspinall, Fiona; Mehta, Raj; Ledger, Jean; Raine, Rosalind; Fulop, Naomi; Barratt, Helen

VERSION 1 – REVIEW

REVIEWER	Cashman, James P Cappagh National Orthopaedic Hospital, Dept of Orthopaedics
REVIEW RETURNED	17-Nov-2022

GENERAL COMMENTS	This is a qualitative assessment of the patient journey of 14 patients undergoing total joint replacement. This is a relatively small volume study in which the authors highlight the difficulties of navigating the waiting list/patient journey in the total joint waiting list. While this paper has some interesting points, I am not sure that the cohort size is reflective of the combined patient experience of all those who undergo joint replacement. This study is at risk of selection bias. I don't think this study significantly adds to our knowledge of the patient journey
---

REVIEWER	Clement, Nicholas Royal Infirmary of Edinburgh, Orthopaedics and Trauma
REVIEW RETURNED	16-Jan-2023

GENERAL COMMENTS	I would like to thank the authors for allowing me to review their work that I am in support of for publication. However, I do feel the authors paint a picture that might be a little worse than actually is experienced. I also suspect following the COVID pandemic and the effect this has had on planned care, with patients likely now waiting years for their joint replacement this should be discussed (as their study was carried out in 2018). The maze will likely seem bigger in view of the longer waits and poorer access to healthcare. From the primary care perspective physio and analgesia is the first line management. To simply put all patients on to a conveyor belt to joint replacement is wrong. I feel this paper supports the more recent work looking at Prehab of patients awaiting arthroplasty - optimising their physical status and informing them of what is to come.
--

REVIEWER	Mehta, Aashna University of Debrecen
REVIEW RETURNED	28-Jan-2023

GENERAL COMMENTS	Thanks for the submission, this is an interesting topic relevant to patient care and overall experience in NHS. While this paper brings out patients' perspectives well, I suggest some changes that can improve the overall quality of the paper:  -The introduction discusses current practice guidelines (line 39 onwards) , it would be great to add some data to bring out its impact on treatment and patient care i.e referral rate, any disparities or impact on patient recovery if noted. -The outcomes should be more clearly defined in the methods section -The discussion fails to discuss the limitations of the study, the limitations should be added.
---

REVIEWER	Lisbôa, Rosa Ladi Universidade Federal de Ciências da Saúde de Porto Alegre, Enfermagem
REVIEW RETURNED	31-Jan-2023

GENERAL COMMENTS	The manuscript ID bmjopen-2022-066762: "Maze not pathway: patients and public experiences of the UK NHS elective total joint arthroplasty pathway" presents elements that configure a relevant topic of interest today regarding the challenge of administration and management of health services, especially, effective care for elective patients. The title reflects the content of the manuscript, although it does not express the methodology/design of the study and the descriptors are sufficient. The abstract is structured and presents all the required items. The article lists five strengths and limitations of the study by the authors. The introduction demonstrates coherence in the presentation of the theme, problem, justification, research question, knowledge gap and logical sequence of ideas, based on the scientific literature, however, it is up to the authors to review the possibility of some substitutions by more recent publications, considering only 28.5% of the references are from the last five years and more than 71% of the literature is from the last five years. The method and result are accurately described, however the discussion, consisting of only three paragraphs, can be better complemented with other findings and foundations from the current literature, so that a discussion really takes place where there is support for ideas. The conclusion item presents the outcome of the research, though, it does not suggest an area for future research. The suggestions and recommendations made were to promote the understanding of the readers and the replication of the study by other researchers in the area.
---

VERSION 1 – AUTHOR RESPONSE

We would like to thank the editor and the reviewers for their comments and suggested changes. We have revised the manuscript considering these comments, grouping the comments thematically and have provided a point-by-point response to explain any changes we have (or have not) made to the original article. Where we have not addressed reviewer comments in the manuscript, we provide additional information here.

Manuscript Section	Comments	Response and action by authors
-----------------	---------------------------------------

Title	Please include the study design in the title. (Editor)	The manuscript title (page 1) has been changed to include the focus group study design.
	The title reflects the content of the manuscript, although it does not express the methodology/design of the study and the descriptors are sufficient. (Reviewer 4)	'Maze' not pathway: focus group exploration of patients' and public experiences of the UK NHS elective total joint arthroplasty pathway.
Abstract	The abstract is structured and presents all the required items. (Reviewer 4)	Thank you for this comment.
Overall	Thanks for the submission, this is an interesting topic relevant to patient care and overall experience in NHS. (Reviewer 3)	Thank you for these comments. As we outline in the methods section, all focus group data were included in thematic analysis. This paper reports the themes participants presented and discussed in relation to their lived experiences of the whole TJA pathway. As the illustrative primary data quotations show, these findings are supported by the data.
	The manuscript ID bmjopen-2022-066762: "Maze not pathway: patients and public experiences of the UK NHS elective total joint arthroplasty pathway" presents elements that configure a relevant topic of interest today regarding the challenge of administration and management of health services, especially, effective care for elective patients. (Reviewer 4)	The authors acknowledge that primary management of joint pain, including physiotherapy and pain management are appropriate and can be the best interventions. Indeed, we acknowledge this in the background section (page 4). However, this paper reports findings from focus group discussions with patients about their experiences of travelling this pathway. The paper describes the lived experience of the entire pathway feeling like a 'conveyor belt' to some of the participants – which was a direct quote from one of the focus group participants (see Quote 21), and many other focus group participants corroborated this experience.
	I would like to thank the authors for allow me to review their work that I am in support of for publication. However, I do feel the authors paint a picture that might be a little worse than actually is experienced. (Reviewer 2)	Further reinforcement of the importance of these approaches to non-surgical management of hip/knee degeneration/pain is now included in the discussion (page 12). We have highlighted that non-surgical interventions (e.g. physiotherapy) prior to surgical referral are perceived by patients and the public as a means of managing demand, despite varying evidence to suggest that prehabilitation is associated with improving some of the outcomes for patients undergoing orthopaedic surgery both preoperatively and postoperatively. While this remains the case, as well as potentially damaging patient trust, it is likely to impact patient engagement with evidence-based care.
	From the primary care prospective physio and analgesia is the first line management. To simply put all patients on to a conveyor belt to joint replacement is wrong.	
Background	While this paper brings out patients' perspectives well, I suggest some changes that can improve the overall quality of the paper: The introduction discusses current practice guidelines (line 39 onwards) , it would be great to add some data to	We have updated the background section (page 4) to include some more information on current practice guidelines – as suggested by Reviewer 3. Due to widespread variation - it has been difficult to get up to date reliable data on referral rates to report in this context for elective orthopaedic surgery. However, the impact on increased wait times due to

	bring out its impact on treatment and patient care i.e. referral rate, any disparities or impact on patient recovery if noted. (Reviewer 3)	conservative management, further exacerbated by the COVID-19 pandemic has been later raised in the discussion section (pages 12-13).
	The introduction demonstrates coherence in the presentation of the theme, problem, justification, research question, knowledge gap and logical sequence of ideas, based on the scientific literature, however, it is up to the authors to review the possibility of some substitutions by more recent publications, considering only 28.5% of the references are from the last five years and more than 71% of the literature is from the last five years. (Reviewer 4)	We have updated some of the references in the introduction in response to other reviewers' comments. However, as the study was grounded in a specific programme – the 'Getting It Right First Time' programme, for which associated publications and reports were launched from 2012 onwards, these are essential for readers to understand the context of the study and this paper.
Methods and Results	The method and result are accurately described (Reviewer 4)	The methods section has been updated with the aim of the research and its primary outcome of concern (which was patient and public perceptions of planned improvements to the NHS TJA pathway) on page 5.
	The outcomes should be more clearly defined in the methods section (Reviewer 3)	
Discussion: Strengths & Limitations	Please discuss the limitations of the study and study design in the Discussion section. (Editor)	We have discussed the study limitations and the study design in the discussion section (page 12) addressing the significance of the contribution to knowledge of the patient journey, including the risk of selection bias and how purposeful sampling was used to mitigate against cohort size (see reference: Sandelowski M. Sample size in qualitative research. Research in Nursing & Health. 1995;18(2):179-183). As this is a qualitative study, it does not follow the same criteria as quantitative research. We specifically explained that focus group methodology was selected to explore a range of views in depth, and not necessarily representation. As participants volunteered to participate, we stated that there may be a degree of self-selection bias, with participants who were more confident in articulating their views and experiences or indeed, those with particular types of experiences, potentially more likely to participate. We used purposeful sampling by contacting 146 community and professional organisations within a reasonable travel distance of
	The article lists five strengths and limitations of the study by the authors. (Reviewer 4)	
	This study is at risk of selection bias. I don't think this study significant adds to our knowledge of the patient journey (Reviewer 1)	
	-The discussion fails to discuss the limitations of the study, the limitations should be added. (Reviewer 3)	
	This is	

	a qualitative assessment of the patient journey of 14 patients undergoing total joint replacement. This is a relatively small volume study in which the authors highlight the difficulties of navigating the waiting list/patient journey in the total joint waiting list. While this paper has some interesting points, I am not sure that the cohort size is reflective of the combined patient experience of all those who undergo joint replacement. (Reviewer 1)	London, UK to reach a range of diverse participants. Target organisations included patient representative groups, local community groups, groups relevant to target participants and the British Orthopaedic Association (BOA) – to ensure we reached a wide range of participants within our eligibility criteria. Specifically, as socio-economic circumstances may impact a patient’s decision to undergo TJA, we explained how we sought to recruit via a wide range of different channels, including professional societies, patient groups, and local community groups (including multi-faith organisations). Nevertheless we were not able to recruit a diverse sample with respect to ethnicity. Indeed, one participant withdrew because of fears of cultural stigma. It is acknowledged that cultural differences may exist between other groups of patient and public members. As participants volunteered to participate, there may be a degree of self-selection bias, with participants who were more confident in articulating their views and experiences or indeed, those with particular types of experiences, potentially more likely to participate. The methodology was selected to explore a range of views in depth, and not necessarily representation. The number of study participants may limit the transferability of findings – although this was mitigated using purposeful sampling there was geographical variability in participants’ lived experiences of the UK TJA pathway.
Discussion	The suggestions and recommendations made were to promote the understanding of the readers and the replication of the study by other researchers in the area. (Reviewer 4) I feel this paper supports the more recent work looking at Prehab of patients awaiting arthroplasty - optimising their physical status and informing them of what is to come. (Reviewer 2) however the discussion, consisting of only three paragraphs, can be better complemented with other findings and foundations from the current literature, so that a discussion really takes place where there	We have included a reference of recent work looking at prehab of patients awaiting joint arthroplasty in the discussion section (page 12). See: Punnoose A, Claydon-Mueller LS, Weiss O, Zhang J, Rushton A, Khanduja V. Prehabilitation for Patients Undergoing Orthopedic Surgery: A Systematic Review and Meta-analysis. JAMA Netw Open. 2023;6(4):e238050. doi:10.1001/jamanetworkopen.2023.8050 We have updated the discussion section (page 12) to include more findings from the current literature to enhance this section. We have also updated the discussion section (page 12) to explain the context of the COVID-19 pandemic on patients’ experiences, and we discuss the likely impact of the study findings following the COVID-19 pandemic. We explained that challenges with traversing the TJA pathway, alongside the negative impacts on physical and mental wellbeing of delayed appointments that were reported in this study, are likely to be exacerbated by the COVID-19 pandemic which has resulted in further delays, cancellations, and increased waiting times for those

	is support for ideas. (Reviewer 4)	on the TJA pathway.
	I also suspect following the COVID pandemic and the effect this has had on planned care, with patients likely now waiting years for their joint replacement this should be discussed (as their study was carried out in 2018). The maze will likely seem bigger in view of the longer waits and poorer access to healthcare. (Reviewer 2)	
Conclusion	The conclusion item presents the outcome of the research, though, it does not suggest an area for future research. (Reviewer 4)	We have included directions for future research in the conclusion (page 13). Research that includes views from both patients and health care professionals is needed at this crucial stages of the TJA pathway to explore how patients are being supported – including the views of both patients and healthcare professionally.

VERSION 2 – REVIEW

REVIEWER	Clement, Nicholas Royal Infirmary of Edinburgh, Orthopaedics and Trauma
REVIEW RETURNED	18-Jun-2023

GENERAL COMMENTS	I would like to thank the authors for considering my, and the other reviewers suggestions, which they have addressed in full. I really did enjoy reading this paper (again).
--

REVIEWER	Mehta , Aashna University of Debrecen
REVIEW RETURNED	04-Jul-2023

GENERAL COMMENTS	Please add limitations pertaining to the study methodology and how they may impact the results obtained. A comparison between the results of group 1,2 and 3 should be illustrated in a table or figure format.
--

REVIEWER	Lisbôa, Rosa Ladi Universidade Federal de Ciências da Saúde de Porto Alegre, Enfermagem
REVIEW RETURNED	15-Jun-2023

GENERAL COMMENTS	Dear authors, I perform the reading and reevaluation of the manuscript. I note that the suggested corrections were fully contemplated or justified by the authors, so I indicate the publication of the manuscript.
---

VERSION 2 – AUTHOR RESPONSE

Reviewer 4 comments	
Dear authors, I perform the reading and re-evaluation of the manuscript. I note that the suggested corrections were fully contemplated or justified by the authors, so I indicate the publication of the manuscript.	Thank you for this comment.
Reviewer 2 comments	
I would like to thank the authors for considering my, and the other reviewers suggestions, which they have addressed in full. I really did enjoy reading this paper (again).	Thank you for this comment.
Reviewer 3 comments	
Please add limitations pertaining to the study methodology and how they may impact the results obtained.	We have updated the limitations section (p12) to incorporate this comment. As previously stated already in the manuscript: The methodology was selected to explore a range of views in depth, and not necessarily representation. We have moved around the ordering of sentences to make this clearer for the reader to understand the intention of selecting this methodology, and its impact on the results.
A comparison between the results of group 1,2 and 3 should be illustrated in a table or figure format.	In writing this manuscript, we have been led by the data and presented this accurately and appropriately throughout the manuscript. We believe it is not appropriate to present this in a tabular format as suggested, as it would imply greater import to the post-hoc analytic comparison and thus we think would be antithetical to the method/purpose.